# Metal-free atom transfer radical polymerization with ppm catalyst loading under sunlight

Qiang Ma[1], Jinshuai Song [2], Xun Zhang[1], Yu Jiang[1], Li Ji[3] & Saihu Liao [1,4,5✉]

Organocatalytic atom transfer radical polymerization (O-ATRP) is recently emerging as an appealing method for the synthesis of metal-free polymer materials with well-defined microstructures and architectures. However, the development of highly effective catalysts that can be employed at a practical low loading are still a challenging task. Herein, we introduce a catalyst design logic based on heteroatom-doping of polycyclic arenes, which leads to the discovery of oxygen-doped anthanthrene (ODA) as highly effective organic photoredox catalysts for O-ATRP. In comparison with known organocatalysts, ODAs feature strong visible-light absorption together with high molar extinction coefficient ($\varepsilon_{455nm}$ up to 23,950 $M^{-1}cm^{-1}$), which allow for the establishment of a controlled polymerization under sunlight at low ppm levels of catalyst loading.

[1] Key Laboratory of Molecule Synthesis and Function Discovery (Fujian Province University), College of Chemistry, Fuzhou University, Fuzhou 350108, China. [2] College of Chemistry and Molecular Engineering, Zhengzhou University, Zhengzhou 450001, China. [3] Grubbs Institute, Southern University of Science and Technology, Shenzhen 518055, China. [4] State Key Laboratory of Photocatalysis on Energy and Environment, College of Chemistry, Fuzhou University, Fuzhou 350108, China. [5] Beijing National Laboratory of Molecular Science (BNLMS), Beijing 100190, China. ✉email: shliao@fzu.edu.cn

Since the discovery in the 1990s, atom transfer radical polymerization (ATRP) has evolved into one of the most versatile and utilized polymerization methods for the synthesis of polymer materials with well-defined structures and architectures, and widely employed in a variety of industrial applications including coatings, adhesives, cosmetics, inkjet printings, etc.[1–3] However, conventional ATRPs rely on transition metal catalysts [i.e., Cu(I), Ru(II)],[3] which will result in transition metal contaminations in the final products, and thus raise concerns when applied to fields sensitive to metal contaminants.[4–6] Therefore, considerable efforts have been dedicated to lowering catalyst loadings or removing residual metals since the initial discovery of ATRP.[5–8] Whereas, the recent emerging organocatalytic atom transfer radical polymerization (O-ATRP) using organic photoredox catalysts, undoubtedly, represents an ideal solution to this challenging issue.[9–15]

Since the conceptual work first demonstrated O-ATRP in 2014 by using organic molecules such as N-phenyl phenothiazine or perylene as a catalyst,[16–18] it has immediately attracted wide research interests in the past 5 years.[19–22] Until now, several frameworks/core structures[22] including phenothiazine (1),[16,23,24] dihydrophenazine (2),[25–28] phenoxazine (3),[29,30] etc.[31–37] have been successfully identified as efficient photocatalysts (PC) for O-ATRP (Fig. 1a). However, a 1000-ppm level of catalyst loading was typically required to reach a satisfactory control over the polymerization. Controlled polymerization at a practical low catalyst loading (<10 ppm) could not only eliminate the need of further product purification or residual catalyst removal, but

### a O-ATRP catalyst development via core structure modification (previous studies)

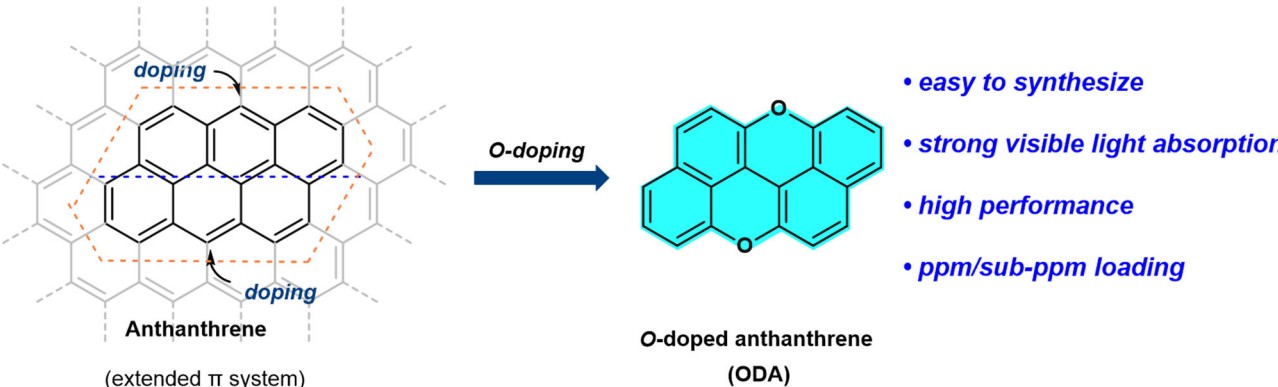

### b O-ATRP catalyst design based on the heteroatom-doping logic (this work)

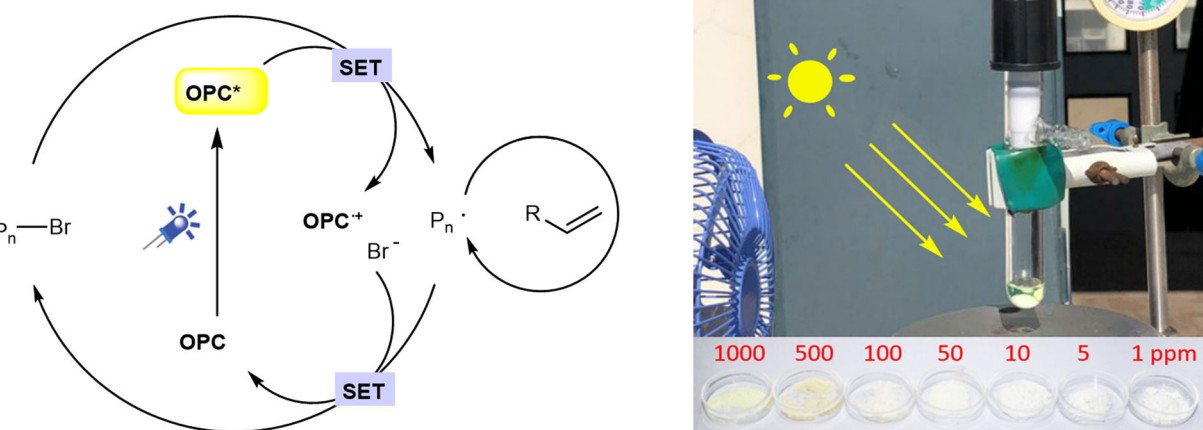

**Fig. 1 The development of catalysts for O-ATRP. a** Catalyst development via core structure modification. **b** O-ATRP photocatalyst design based on a heteroatom-doping logic (this work). **c** Catalytic cycle for a light-mediated O-ATRP. **d** Polymerization under sunlight and the product color (PMMA). *SET* single electron transfer, *OPC* organic photocatalyst, *PMMA* polymethyl methacrylate.

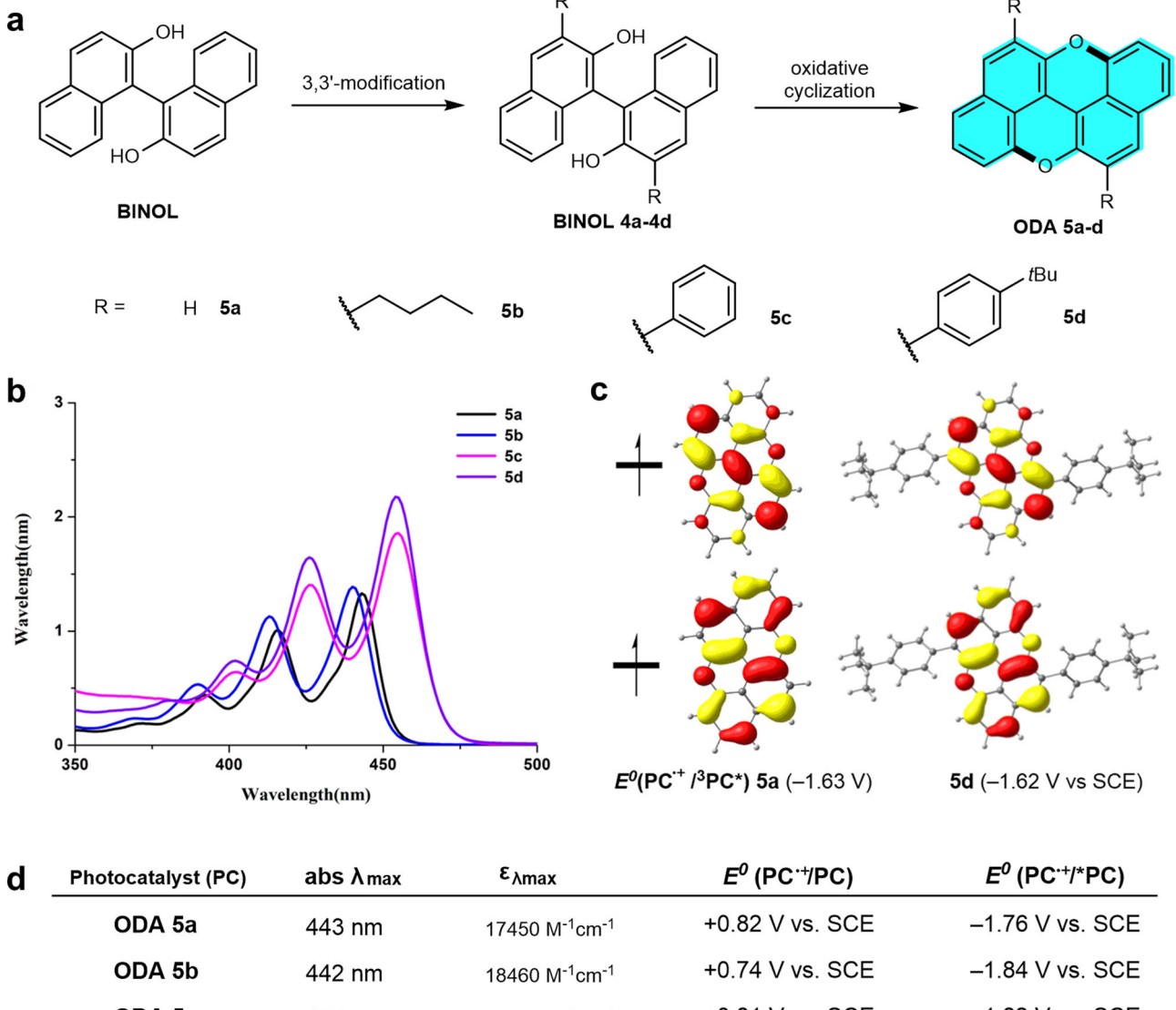

**Fig. 2 Synthesis and characterization of photocatalysts. a** Synthesis of oxygen-dopants of anthanthrene **5a–d**. **b** UV-Vis absorption profiles. **c** SOMO orbitals and triplet reducing power of **5a** and **5d**. **d** Characterization data of photocatalyst **5a–5d** by UV-Vis, fluorescence emission, CV, and calculated redox potentials. *SCE* saturated calomel electrode, *SOMO* singly occupied molecular orbital, *CV* cyclic voltammetry.

| Photocatalyst (PC) | abs $\lambda_{max}$ | $\varepsilon_{\lambda max}$ | $E^0$ (PC$^{·+}$/PC) | $E^0$ (PC$^{·+}$/*PC) |
|---|---|---|---|---|
| **ODA 5a** | 443 nm | 17450 M$^{-1}$cm$^{-1}$ | +0.82 V vs. SCE | −1.76 V vs. SCE |
| **ODA 5b** | 442 nm | 18460 M$^{-1}$cm$^{-1}$ | +0.74 V vs. SCE | −1.84 V vs. SCE |
| **ODA 5c** | 454 nm | 22580 M$^{-1}$cm$^{-1}$ | +0.81 V vs. SCE | −1.82 V vs. SCE |
| **ODA 5d** | 455 nm | 23950 M$^{-1}$cm$^{-1}$ | +0.80 V vs. SCE | −1.84 V vs. SCE |

could also decrease the cost of commercial production.[14,28] Therefore, the development of highly effective photocatalysts for O-ATRP has thus become a focus of extensive studies in recent years.[20–22,24,28] Whereas, many mechanistic aspects in O-ATRP remain unclear so far, and there is still a lack of general guidelines for the catalyst design.[22,38]

Phenothiazine,[16] dihydrophenazine,[25] and phenoxazine[29] are among the most efficient and widely used catalyst frameworks that could achieve a well-controlled O-ATRP polymerization with low dispersity.[22] To further increase the catalyst efficiency, much effort has been dedicated to the modifications of the catalyst structures such as introducing aryl groups, which could enhance the visible-light absorption, but the red shift of absorption maximum ($\lambda_{max}$) was often quite limited.[21,22] Interestingly, an analysis on the UV-visible-light absorption of these photocatalysts unveiled a uniformly decreased light absorption profile from its absorption maximum (<400 nm) to visible light.[16,22,25,29] Therefore, we questioned whether there is a possibility to find a suitable and tunable O-ATRP catalyst framework with its

absorption maximum located in the visible-light region, and a stronger light absorption could probably lower the catalyst loading. The three photocatalysts (**1–3**) can be recognized as derivatives obtained by modifying the corresponding core chromophoric structures: phenothiazine, dihydrophenazine, and phenoxazine, respectively (Fig. 1a). Recently, we were thinking that heteroatom-doping[39,40] of small polycyclic arenes might be a feasible catalyst design logic for the O-ATRP photocatalyst development (Fig. 1b).

According to the oxidative quenching mechanism (Fig. 1c), organic photoredox catalysts possessing a highly reducing excited state are required to reduce the alkyl bromides via a single electron transfer (SET) to initiate the polymerization.[41] Besides, redox potential, photophysical property, stability, etc. of the related catalytic species are also critical to establish a fast and effective switching between propagating and dormant states of the macro-initiators, thus achieving a controlled polymerization with narrow dispersities.[38,41–43] The integration of heteroatoms such as N, O, S, etc. into aromatic hydrocarbons could regulate

the photophysical and photoredox properties of these polycyclic arene catalysts[39,41], and thus probably lead to an improvement in their catalytic performance on polymerization and suppress undesired catalyst decomposition or catalyst-initiation.[17,31,32] With this idea in mind, we thus decided to practice this heteroatom-doping strategy on graphene, and narrowed down the parental aromatic (extended π) system to anthanthrene (Fig. 1b, left), which possesses the desired strong absorption in the visible-light region.[44] Oxygen-doping was chosen for the current research, as: (i) the known high-performance photocatalysts for O-ATRP were developed based on the charge-transfer (CT) principle,[24–30] and typically contain a triaryl amine part, which is required to impose a twisted donor–acceptor structure to favor the CT process.[24–28,45] Therefore, the development of a completely new photocatalyst framework lacking this moiety and that could also achieve the same level or even better performance can be of fundamental significance, which, to the best of our knowledge, remains unknown so far; (ii) O-doping can regulate the photophysical and redox properties of anthanthrene, probably leading to a more oxidizing $OPC^{\bullet+}$ in comparison with the typical amine-containing catalysts 1–3, which may thus afford a better deactivation control.[25–28,41] Here, we report our efforts toward this goal, and the discovery of oxygen-doped anthanthrene (ODA) as an effective organocatalyst framework for O-ATRP, which exhibits strong absorption at the visible-light region ($\varepsilon_{450nm} > 20,000\ M^{-1}\ cm^{-1}$), and allows for a ppm/sub-ppm level of catalyst loading only to deliver a metal-free, controlled polymerization under visible light or even sunlight (Fig. 1d).

## Results

**Catalyst synthesis and characterization.** The ODA can be readily constructed via dual oxidative cyclization[46] from the commercially available 1,1′-bisnaphthol (BINOL) (Fig. 2a). Further, by virtue of its high modifiability,[47] ODAs with different substituents can be readily synthesized via a 3,3′-modification of BINOL/oxidative cyclization sequence (see Supplementary Methods). To improve the catalyst solubility, we synthesized the n-butyl substituted catalyst 5b, while 5c and 5d were prepared to examine the influence of aryl substituents (for synthetic procedures and spectra, see Supplementary Information). As anticipated, 5a did show a strong absorption in the visible-light region (Fig. 2b), with a red shift of the absorption maximum from 437 to 443 nm (comparing with the parental anthanthrene[44]). This absorption profile is in sharp

contrast to that of the known photocatalysts such as 1–3, which possess a smaller conjugation in the core structure and absorption maximums appearing in the ultra-violet region (<400 nm). Both the absorption maximums ($\lambda_{max} = 454$ and 455 nm) of 5c and 5d have shown a further red shift (ca. 12 nm) as well as enhanced light absorption ($\varepsilon_{max} = 22,580$ and 23,950 vs 17,450 $M^{-1}\ cm^{-1}$). In addition, a notable feature of the ODA catalysts substantially different from the known high-performance O-ATRP photocatalysts[24,28] is the lack of a charge-separation in their excited states (Fig. 2c). Based on the fluorescence emission and the cyclic voltammetry (CV) data, we could assess the reducing capability of the excited singlet states ($E^{0*}(PC^{\bullet+}/^1PC^*)$) of these catalysts, ranging from −1.76 to −1.84 V vs SCE (saturated calomel electrode) (Fig. 2d). Density functional theory (DFT) was used to estimate the reduction ability of the triplet excited-states ($E^0$ ($PC^{\bullet+}/^3PC^*$), which are in the range of −1.58 to −1.71 V vs SCE (Fig 2c, for details, see Supplementary Information and Computational details). Although the excited states of ODA 5a–5d are less reductive than that of N-phenyl phenothiazine (1, Ar = Ph, R = H, $E^{0*} = -2.1$ V vs SCE),[16] all are more negative than −0.7 V that is required to reduce ethyl α-bromophenylacetate (EBP, a common initiator, $E^0(EBP/EBP^{\bullet-}) = -0.74$ V SCE).[25] Notably, the radical cations ($PC^{\bullet+}$) of ODA catalysts are much more oxidizing (up to +0.82 V vs SCE), in comparison with the most efficient dihydrophenazine-based photocatalyst (+0.38 V vs SCE).[28]

**Initial evaluation of the photocatalysts.** We conducted the initial evaluation of these oxygen-doped catalysts in the polymerization of methyl methacrylate (MMA) by using EBP as initiator, dimethylacetamide (DMA) as solvent under the irradiation of purple light-emitting diodes (LEDs, $\lambda_{max}$ 400 nm). To our delight, 5a could afford a controlled polymerization with a moderate dispersity (Table 1, entry 1). The polymerization can also be conducted with control in other solvents, such as dimethylformamide (DMF, entry 2), dichloromethane (DCM), toluene, tetrahydrofuran, etc. (Supplementary Table 2). DCM gave the lowest dispersity (Đ = 1.25) together with a good agreement between the experimental and theoretical $M_n$ (entry 3). Remarkably, ODA 5a is also effective with other initiators such as ethyl-bromopropanoate (EBrP), ethyl-2-bromoisobutanoate (EBiB), diethyl 2-bromomalonate (DBM), and diethyl 2-bromo-2-methylmalonate (DBMM) (Đ 1.19–1.27, entries 4–7). 5b–5d with different substituents were then compared under the standard conditions by using EBP as the initiator. Pleasingly, all the three new catalysts are effective for the polymerization (entries

**Table 1 Catalyst evaluation in the atom-transfer radical polymerization of MMA.**

| Entry | PC | Initiator | Solvent | Conv. (%) | $M_{n,\ theo}$ (kDa) | $M_{n,\ GPC}$ (kDa) | Đ |
|---|---|---|---|---|---|---|---|
| 1 | 5a | EBP | DMA | 88.5 | 9.10 | 19.9 | 1.34 |
| 2 | 5a | EBP | DMF | 83.4 | 8.59 | 18.3 | 1.33 |
| 3 | 5a | EBP | DCM | 73.3 | 7.58 | 12.8 | 1.25 |
| 4 | 5a | EBrP | DCM | 72.7 | 7.46 | 13.3 | 1.27 |
| 5 | 5a | EBiB | DCM | 78.2 | 8.02 | 14.9 | 1.23 |
| 6 | 5a | DBM | DCM | 80.8 | 8.33 | 13.7 | 1.22 |
| 7 | 5a | DBMM | DCM | 81.2 | 8.38 | 12.8 | 1.19 |
| 8 | 5b | EBP | DCM | 66.5 | 6.90 | 12.3 | 1.19 |
| 9 | 5c | EBP | DCM | 79.2 | 8.17 | 13.8 | 1.22 |
| 10 | 5d | EBP | DCM | 88.2 | 9.07 | 13.5 | 1.23 |
| 11 | 5b | DBMM | DCM | 69.1 | 7.17 | 13.5 | 1.15 |
| 12 | 5d | DBMM | DCM | 84.8 | 8.75 | 12.0 | 1.12 |
| 13[a] | 5a | EBP | DCM | 82.7 | 8.52 | 15.7 | 1.25 |
| 14[a] | 5d | DBMM | DCM | 71.8 | 7.44 | 10.9 | 1.15 |

Reaction conditions: [MMA]$_0$:[initiator]$_0$:[**PC**]$_0$ = 100:1:0.05, solvent (9.4 M of MMA), at room temperature, under the irradiation of purple LEDs (400 nm, 25 mW cm$^{-2}$), 10 h. Conversions were determined by $^1$H NMR. $M_n$ and Đ were determined by GPC with polymethyl methacrylate (PMMA) standards. *Abbreviations: MMA* methyl methacrylate, *PC* photocatalyst, *Conv.* conversion, *GPC* gel permeation chromatography, Đ = $M_w/M_n$, *EBP* ethyl α-bromophenylacetate, *EBrP* ethyl-bromopropanoate, *EBiB* ethyl-2-bromoisobutanoate, *DBM* diethyl 2-bromomalonate, *DBMM* diethyl 2-bromo-2-methylmalonate, *DMA* dimethylacetamide, *DMF* dimethylformamide, *DCM* dichloromethane. [a]Irradiated by blue LEDs (460 nm, 30 mW cm$^{-2}$), 8 h.

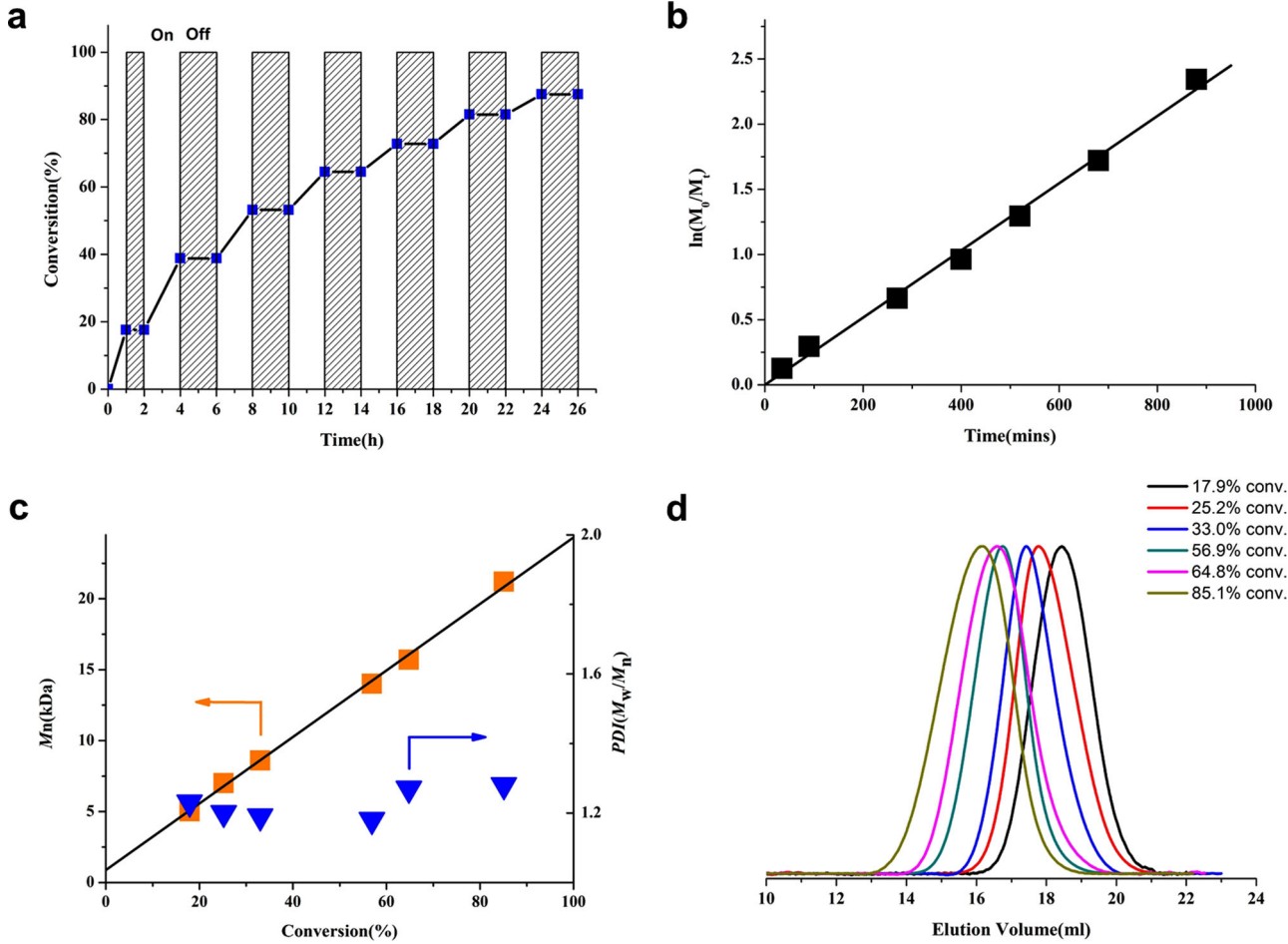

**Fig. 3 Temporal control and kinetic study on ODA 5d-catalyzed ATRP of MMA. a** Light on–off experiments and the plot of monomer conversion vs time. **b** Kinetic plot for the metal-free ATRP. **c** Plot of $M_n$ and Đ vs monomer conversion for the polymerization of MMA under continuous irradiation. **d** GPC traces of each polymer depicted in (**c**) (color coded). All polymerizations were performed at a ratio of $[MMA]_0:[DBMM]_0:[\mathbf{5d}]_0 = 200:1:0.05$ under purple LED irradiation (400 nm, 25 mW cm$^{-2}$).

8–10), and ODA **5b** could achieve a dispersity (Đ) lower than 1.20 (entry 8). Catalysts **5b** and **5d** were further examined with DBMM (entries 12 & 13), and a remarkable narrow dispersity (Đ = 1.12, entry 12) was obtained with **5d**. Irradiation with lower energy blue LEDs was also effective (entries 13 & 14).

**Light regulation, kinetics, and block polymer synthesis.** A prominent feature of the O-ATRP is that the polymerization can be regulated by light.[48,49] To examine the temporal control ability of the current system, a light on–off experiment was performed with light on–off cycle repeating for several times until over 90% conversion was achieved. As depicted in Fig. 3a, the polymerization only preceded in the presence of light irradiation, while no polymerization was observed in dark. Further, light on–off experiments have shown no conversion even over a long dark period up to 12 h. This strict control by light over the whole process suggests an effective activation and deactivation mechanism of the polymerization. The polymerization was then followed by [1]H NMR to gain some insights about the polymerization kinetics, which unveiled a first-order kinetics through the course of the reaction (Fig. 3b). The $M_{n,GPC}$ values were plotted against the monomer conversion, as shown in Fig. 3c, and a linear increase of molecular mass throughout the polymerization process was observed. Notably, the y-intercept of the $M_n$ vs conversion plot was 870 Da, indicating that the control of the

polymerization was achieved after the initial addition of ~6 MMA (Fig. 3c).[25] This strict control may benefit from a more efficient deactivation process[26,37,45] due to the radical cations of ODA photocatalysts possessing a better oxidizing ability, when compared with the dihydrophenazine-derived photocatalysts.[25,28]

Regarding the chain-end fidelity, a polymethyl methacrylate (PMMA) sample obtained via **5d**-mediated metal-free ATRP was subjected to MALDI-TOF (matrix-assisted laser desorption/ ionization time-of-flight mass-spectrometer) analysis (Supplementary Fig. 10), which shows a consistence observed molecular weight with the expected values with each peak separated by the mass of one MMA (100 Da), and individual PMMA polymers having the initiating unit at one chain end and a bromine atom at the propagating chain end. An advantage of high chain-end fidelity is it enables the synthesis of block polymers. As shown in Fig. 4, chain extension (PMMA-b-PMMA) and block copolymerization products (PMMA-b-PBnMA and PMMA-b-PBA) can all be prepared with this catalytic system. Notably, the gel permeation chromatography (GPC) traces, all clearly show an obvious shift to higher molecular weight species with little tailing in the homo-polymer regime, giving further support to the high alkyl bromide chain-end fidelity in the PMMA macro-initiators (purified via re-precipitation from methanol) and also a high re-initiation efficiency. Of note, this result is also in consistence with the high initiator efficiency observed in the polymerizations with freshly distilled DBMM (Supplementary Table 8). Moreover,

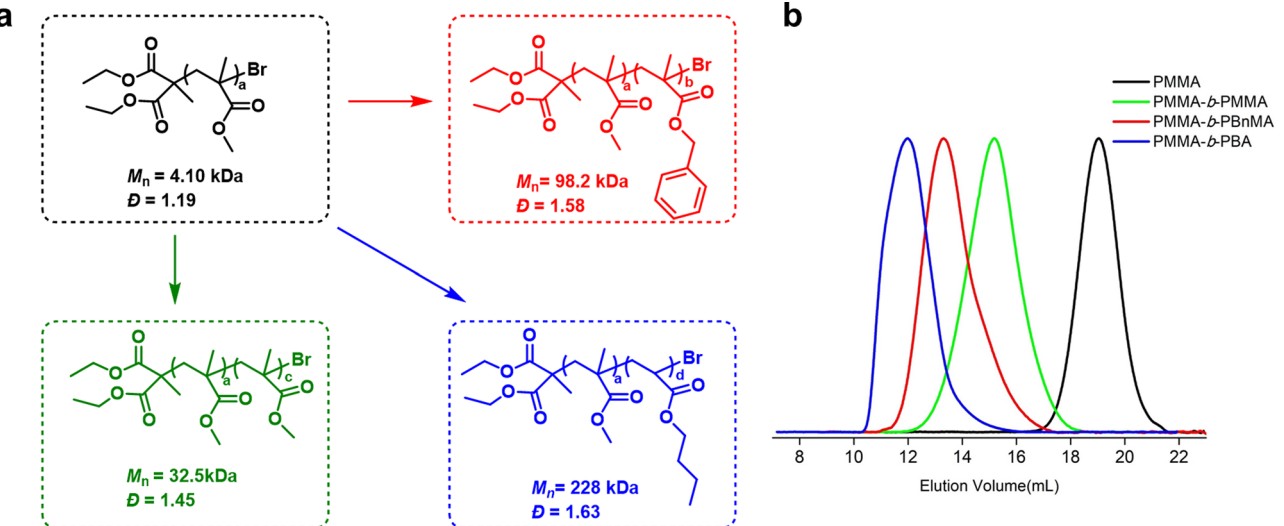

**Fig. 4 Block polymer preparation. a** Chain-extension from a PMMA macro-initiator (black) to produce block copolymers after a further polymerization with MMA (green), BnMA (red), and BA (blue). **b** GPC traces of the corresponding polymers depicted in left by using the same color coded. *BnMA* benzyl methacrylate, *BA* butyl acrylate.

**Table 2 Polymerization with low catalyst loadings.**

| Entry | PC loading (ppm) | Monomer | Initiator | PC | Light source | Time (h) | Conv. (%) | $M_{n,GPC}$ | Đ |
|---|---|---|---|---|---|---|---|---|---|
| 1 | 100 | MMA | EBP | **5a** | Purple LEDs | 10 | 71.4 | 14.3 | 1.19 |
| 2 | 10 | MMA | DBMM | **5a** | Blue LEDs | 10 | 88.1 | 14.8 | 1.20 (1.09) |
| 3 | 10 | MMA | DBMM | **5d** | Blue LEDs | 12 | 90.4 | 14.5 | 1.17 (1.08) |
| 4 | 5 | MMA | DBMM | **5d** | Blue LEDs | 12 | 83.2 | 14.3 | 1.25 |
| 5 | 0.5 | MMA | DBMM | **5d** | Blue LEDs | 14 | 77.4 | 19.2 | 1.34 (1.26) |
| 6 | 0.1 | MMA | DBMM | **5d** | Blue LEDs | 14 | 70.3 | 25.6 | 1.39 |
| 7 | 0.05 | MMA | DBMM | **5d** | Blue LEDs | 14 | 62.5 | 30.8 | 1.51 |
| 8[a] | 10 | MMA | DBMM | **5d** | Sunlight‡ | 7 | 50.7 | 11.4 | 1.31 |
| 9[a] | 50 | MMA | DBMM | **5d** | Sunlight‡ | 7 | 49.2 | 9.40 | 1.22 |
| 10 | 10 | TFEMA | DBMM | **5d** | Blue LEDs | 12 | 87.5 | 13.6 | 1.22 |
| 11 | 50 | TFEMA | DBMM | **5d** | Blue LEDs | 12 | 81.6 | 11.5 | 1.09 |
| 12 | 50 | BnMA | DBMM | **5d** | Blue LEDs | 12 | 93.5 | 11.5 | 1.29 |
| 13 | 10 | BA | DBMM | **5d** | Blue LEDs | 7 | 78.2 | 22.3 | 1.49 (1.37) |
| 14 | 50 | BA | DBMM | **5d** | Blue LEDs | 7 | 91.6 | 30.8 | 1.41 (1.29) |

The polymerizations were performed under standard conditions. Conversions were measured by ¹H NMR. $M_n$ and Đ were determined using GPC with PMMA standards. Đ in parentheses were determined using GPC coupled with multi-angle light scattering (MALS). Purple LEDs ($\lambda_{max}$ 400 nm, 25 mW cm⁻²); Blue LEDs ($\lambda_{max}$ 460 nm, 30 mW cm⁻²). *Abbreviations*: PC photocatalyst, *Conv.* conversion, *GPC* gel permeation chromatography, $Đ = M_w/M_n$, *MMA* methyl methacrylate, *TFEMA* 2,2,2-trifluoroethyl methacrylate, *BnMA* benzyl methacrylate, *BA* butyl acrylate, *EBP* ethyl α-bromophenylacetate, *DBMM* diethyl 2-bromo-2-methylmalonate. ᵃSunlight experiments were performed outside on a sunny winter day at Fuzhou University. ‡(26 °N, 119 °E, 25 °C).

triblock copolymer synthesis is also viable with this system as demonstrated with the preparation of PMMA-*b*-PBnMA-*b*-PBA (see Supplementary Fig. 13).

**Polymerization at low ppm catalyst loadings**. The high molar-extinction coefficient ($\varepsilon_{max}$ upto 23,950 M⁻¹ cm⁻¹) of ODA catalysts at the visible-light region encouraged us to examine their performance at low catalyst loadings. As shown in Tables 1 and 2, decreasing the catalyst loading from 500 to 10 ppm, both photocatalysts **5a** and **5d** could maintain their catalytic efficiency, giving a controlled polymerization with low dispersity (Table 2, entries 1–3). To our delight, in the present system, very narrow dispersity can be achieved with 10 ppm **5d** only (entry 3). Remarkably, the catalyst loading of **5d** can be further decreased from 10 to 0.1 ppm (entries 3–6), and even to 50 ppb by more than two orders of magnitude (entry 7), which is among the lowest catalyst loadings so far in O-ATRP.[28] It is worth mentioning that a catalyst loading less than 10 ppm could be very meaningful, which may eliminate the need of the catalyst removal

process. Importantly, white polymer products can be obtained at the catalyst loading less than 50 ppm, even though the photocatalysts are normally colored compounds (Fig. 1d). The high catalytic performance may benefit from their strong visible-light absorption (**5d**, $\varepsilon_{455nm} = 23,950$ M⁻¹ cm⁻¹). In fact, the strength of visible-light absorption is approximately one order of magnitude stronger than that of *fac*-Ir(ppy)₃ ($\varepsilon_{458nm} = 2450$ M⁻¹ cm⁻¹), which is a commonly used organometallic photoredox catalyst.[50] Remarkably, sunlight is also suitable to drive the polymerization with 10 ppm catalyst only (entries 8 & 9). To the best of our knowledge, 10 ppm represents the lowest catalyst loading achieved so far in controlled O-ATRP under sunlight. Furthermore, other methacrylate monomers such as 2,2,2-trifluoroethyl methacrylate (TFEMA) and benzyl methacrylate (BnMA) can also be polymerized with low dispersity at 10 or 50 ppm catalyst loading (entries 10–12). Surprisingly, ODA catalysts could also deliver a controlled polymerization with low dispersity in the polymerization of *n*-butyl acrylate (entries 13 & 14, also see Supplementary Table 6). It is worth mentioning that acrylate

monomers proved to be very challenging to achieve a controlled polymerization with low dispersity by O-ATRP methods due to their high polymerization rate.[25,28,45] To our delight, this control on low dispersity is even better than that with dihydroacridine photocatalysts, which was developed especially for this type of monomers very recently.[45] Notably, ODA **5d** also represents a rare example of organic photocatalysts that could promote the polymerization of both MMA and BA with good controls.[28,45]

## Discussion

In this work, a catalyst design logic based on heteroatom-doping of polycyclic arenes has been successfully introduced into the development of photocatalysts for O-ATRP. ODA has been identified as a catalyst framework with high efficiency, featuring its strong visible-light absorption and excellent performance at very low catalyst loadings (50 ppb to 10 ppm) in organocatalytic ATRP. This framework also represents a rare example of non-*N*-heterocyclic organocatalysts lacking a CT character that could mediate the ATRP with good control and low dispersity (Đ < 1.20) at a ppm level of catalyst loading. We anticipate that this class of photoredox catalysts will find further applications in polymer synthesis and other photocatalysis-related fields.

## Methods

**Typical procedure for O-ATRP of MMA under light**. Typical metal-free organocatalytic ATRP procedures with the molar ratio of [MMA]$_0$:[initiator]$_0$:[catalyst]$_0$ = 100:1:0.05 were shown as follows. The polymerization was conducted with MMA (1.0 mL, 9.35 mmol, 100 eq.) as the model monomer, DBMM (18 μL, 93.5 μmol, 1.0 eq.) as the ATRP initiator, organic photocatalyst (4.70 μmol, 0.5 eq.), and DCM (1.0 mL) as the solvent in a Schlenk tube with a PTFE stirring bar. The mixture was deoxygenated by freeze–pump–thaw cycle three times, backfilled argon, and sealed up subsequently. And then the polymerization occurred under purple LED or blue LED or sunlight irradiation at room temperature. After the desired time, the tube was opened under argon and 20.0 μL of mixture was syringed out and quenched into CDCl$_3$ containing 250 ppm BHT to determine the monomer conversion by $^1$H NMR. The reaction mixture was then diluted with 0.5 mL dichloromethane and dissolved completely, then dripped into 75 mL methanol and stirred for 2 h. The precipitate was then collected by suction filtration with a Buchner funnel and dried in vacuum oven until a constant weight was achieved, at 30 °C to give the purified polymers. For details, additional data, and experiments, please see the Supplementary Information (Supplementary Methods, Supplementary Figs. 1–18, Supplementary Tables 1–8, etc.).

## Data availability

The authors declare that all data supporting the findings of this study are available within the article and Supplementary Information files, and are also available from the corresponding author upon reasonable request.

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

## Acknowledgements

We thank Prof. Krzysztof Matyjaszewski (Carnegie Mellon University) and Prof. Junfang Li (Shanghai Institute of Organic Chemistry, CAS) for helpful discussions and suggestions. Dr. Zechun Jiang is highly appreciated for his support with GPC (Wyatt MALS) measurements. Many thanks to Dr. Zhiwei Lin (Xiamen University) for her help in the MALDI-TOF test, and to Prof. Jianjun Sun and Ms. Xingyuan Du (Fuzhou University) for their help in electrochemical measurements. We thank the Recruitment Program of Global Experts, National Natural Science Foundation of China (21602028), Beijing National Laboratory for Molecular Sciences (BNLMS201913), the Hundred-Talent Project of Fujian, and Fuzhou University for the financial support. In memory of my beloved son, David Liao, who remains a lifelong inspiration for my scientific work (S.L.).

## Author contributions

Q.M. conducted the research on the catalyst synthesis, polymerization evaluation, kinetic study, characterization, etc.; J.S. performed the computational study; X.Z., Y. J., and L.J. participated in the catalyst synthesis and polymerization development; S.L. conceived this concept and prepared this manuscript with feedback from Q.M. and J.S.

## Competing interests

The authors declare no competing interests.
