## [Peer Review File · Nature Communications]

REVIEWER COMMENTS

Reviewer #1 (Remarks to the Author):

This is a very interesting article which presents the use of heterocyclic arene to activate an O-ATRP system (oxygen doping anthracene). The paper is well organized and the data support the conclusions. In my opinion, this paper reports a significant finding which deserves publication in Nature Communication.

I have very minor comments

1. in the introduction, the authors justify their work by the potential toxicity of copper or Ru. However, they did not provide clear demonstration that their catalysts is not toxic. I suggest to amend it or to show.
2. How stable are these catalysts (photobleaching)?
3. It will be also interesting if the authors could comment on the difference between the theoretical and experimental molecular weight values. it looks that in DMAc and DMF, the difference is significant. However, in DCM, this difference is less significant. I am curious to know if DCM transfers during the ATRP polymerization.
4. Figure 3C displays Mn versus monomer conversion. it looks that the exp and theoretical values align well; but this is in slight contradiction of Table 1. can the authors comment on this? also the dispersity values tend to be around 1.2-1.3. Can the addition of PC can reduce this value?
5. It will be interesting if the authors could provide a long off in the main text.
6. for chain extension, i suggest to plot MWD (not retention time) it will be clearer.
7. light intensity should be included in the captions of the figures
8. The authors may cite the following papers: a comprehensive review: Copper-mediated living radical polymerization (atom transfer radical polymerization and copper (0) mediated polymerization): from fundamentals to bioapplications, Chemical reviews 116 (4), 1803-1949
N,N-Diaryl Dihydrophenazines as Photoredox Catalysts for PET-RAFT and Sequential PET-RAFT/O-ATRP
ACS macro letters 7 (6), 662-666

In conclusion, in my opinion, it is a strong study which deserves publication. The work is significant and new. i think it will have a significant impact in the polymer field

Reviewer #2 (Remarks to the Author):

This manuscript reports the discovery of oxygen-doped antratherene as a new class of organic photoredox catalysts for metal-free ATRP. The reported catalyst features strong visible light absorption as well as good control over the whole polymerization process so that only low ppm levels of the catalyst was required to achieve the narrow molecular weight distributions. The most interesting part in this work is that the author rationalized the choice of oxygen doping in antratherene, leading to a superb catalyst in metal-free ATRP. The manuscript was well organized and presented, and I would like to recommend to publish this work after addressing several small points.

1. In table 1, DCM was used as a solvent in entry 3-14. In my knowledge, DCM is not a good solvent in ATRP since it could initiate the polymerization under ATRP conditions.
2. In table 2, the columns of PC and initiator were shifted. For entry 15-17, CPADB was used as the initiator, so these reactions were not classified as ATRP but RAFT. It might be good to remove these examples.
3. For almost all the examples, the obtained experimental Mn were about 50% more than theoretical values, which indicates a lower initiator efficiency. However, the dispersitis are very low. The polymer sample should not be purified, and they should be directly analyzed by GPC to see the real Mn.
4. It would be better to cite a recent review of ATRP: Chem. Soc. Rev., 2018, 47, 5457-5490.

Reviewer #3 (Remarks to the Author):

Dear Authors

The studies developed in this work evaluates a catalyst design based on heteroatom-doping of polycyclic arenes as a new class of organic photoredox catalysts for O-ATRP.

I agree that this topic is of paramount importance, and believe that computationally speaking, the methodology employed in this work is sound and thus the results should be trustworthy. However minors revision is required before the paper can be considered for publication in Nature Communications. Below are the comments of the reviewer:

1) Define the acronyms the first time they appear in the text

2) Review the English. I suggest making the text clearer.

Define all terms, abbreviations, acronyms mainly in the tables.

3) I suggest arranging the numerical order of the Figures in the manuscript. Figure 1a, Figure 1b, Figure 3c Example, figure 1c cannot appear before 1b. This is valid for every manuscript

About the theoretical part:

4) Why did the authors use PBE0?

5) It was not indicated whether negative frequencies appeared. I suggest adding this information.

6) To determine E_0 (PC+/3PC*) (Supl. Mat.) the authors used two reference values SHE (-4.48V) and to SCE (-0.244V) in acetonitrile. Indicate why of the values and why you use them.

7) Finally, I believe that TD-DFT calculations can bring more information to the paper. I suggest performing calculations at the TD-DFT level and correlating the results with the experimental data.

Best regards,
Douglas Henrique Pereira

Dear Reviewers,

Many thanks for your insightful comments and helpful suggestions, which allowed a further improvement of our research and also the presentation of this work. We have carefully revised the manuscript and supplementary information accordingly. Thank you very much.

My best regards

Sincerely

Saihu Liao

Point-by-point responses:

Reviewer #1 (Remarks to the Author):

This is a very interesting article which presents the use of heterocyclic arene to activate an O-ATRP system (oxygen doping anthracene). The paper is well organized and the data support the conclusions. In my opinion, this paper reports a significant finding which deserves publication in Nature Communication.

I have very minor comments

1. In the introduction, the authors justify their work by the potential toxicity of copper or Ru. However, they did not provide clear demonstration that their catalysts is not toxic. I suggest to amend it or to show.

Response: Thank you very much for the comments and suggestion. As suggested, we have removed that sentence in the revised manuscript.

2. How stable are these catalysts (photobleaching)?

Response: Till the end of polymerization, we did not observe obvious color changes (please see the photos shown below). Further, when considering the low catalyst loadings (e.g. 50 ppb) and 10 ppm catalyst loading in block copolymer synthesis, the ODA catalysts should be quite stable.

3. It will be also interesting if the authors could comment on the difference between the theoretical and experimental molecular weight values. It looks that in DMAc and DMF, the difference is significant. However, in DCM, this difference is less significant. I am curious to know if DCM transfers during the ATRP polymerization.

Response: The better agreement between theoretical and experimental molecular weights observed in DCM may benefit from the better solubility of the photocatalyst (**5a**, 500 ppm) in DCM. In fact, at 500 ppm catalyst loadings, we could see some insoluble solids suspended in the reaction system with DMF and DMAc as solvent.

The initiation by DCM should be negligible, as: 1) when the polymerization carried out in DCM in the absence of external initiators, almost no polymerization occurred (<5% conversion, please see Supplementary Table 4). Of note, this result also suggests that the photocatalyst-initiation should be negligible; 2) In contrast to the quantitative (>98%) yield in the ATRA (atom-transfer radical addition) reaction of bromine initiator (diethyl 2-bromomalonate, DBM) with olefin using ODA catalyst (shown below), no ATRA product (no reaction) was detected by GC-MS analysis when dichloromethane was used instead of DBM.

Furthermore, we carried out the polymerization at 10 ppm catalyst loading in different solvents (see the table below). As expected, we could observed better agreement between $M_n(\text{theo})$ & $M_n(\text{GPC})$ in all cases (DMA, DCM, DMF, & Toluene, I^* 75-90%), with toluene (entry 4), which is the best solvent to dissolve the photocatalyst, giving the highest agreement. We have included these results in Supplementary Table 7 in the revised Supplementary Information.

Entry	Photocatalyst	Solvent	Conv	M_n (kDa)	\bar{D}	I^* %
1	5d	DMA	81.5%	11.2	1.25	75.2
2	5d	DCM	96.9%	12.2	1.19	82.5
3	5d	DMF	87.7%	11.8	1.28	76.9
4	5d	Toluene	92.5%	10.6	1.34	89.5
5	5d	THF	88.2%	11.7	1.34	77.2

Reaction conditions: [MMA]:[DBMM]:[5d]= 100:1:0.001 (10 ppm) with freshly distilled DBMM, blue LEDs.

4. Figure 3C displays M_n versus monomer conversion. it looks that the exp and theoretical values align well; but this is in slight contradiction of Table 1. can the authors comment on this? also the dispersity values tend to be around 1.2-1.3. Can the addition of PC can reduce this value?

Response: Figure 3C are results of polymerization at the monomer/initiator ratio of 200/1, while the results shown in Table 1 were obtained at monomer/initiator ratio of 100/1. To some extent, increasing the catalyst loading (10 ppm to 500 ppm) can further (slightly) reduce the dispersity values (see Table 1, entry 7 & 12 vs Table 2, entry 2 & 3, also see Supplementary Table 8, entry 6 vs 7). Thank you for the suggestion.

5. It will be interesting if the authors could provide a long off in the main text.

Response: Thank you very much for the suggestion. The light temporal control is an important part in our study. We checked the polymerization before with a long off up to 12 hours, which also shown no conversion in dark and the polymerization can be re-initiated when exposed to light again. We have added a comment on the long off experiment in the revised manuscript (Page 8, Line 5).

6. for chain extension, i suggest to plot MWD (not retention time) it will be clearer.

Response: We contacted the engineers of Waters Company, but they said we could not export these data showing MWD with our GPC. Thank you for your understanding.

7. Light intensity should be included in the captions of the figures

Response: Thank you very much for the suggestion. We have added the light intensity in the captions of the figures and tables in the revised manuscript.

8. The authors may cite the following papers: a comprehensive review: Copper-mediated living radical polymerization (atom transfer radical polymerization and copper (0) mediated polymerization): from fundamentals to bioapplications, *Chemical reviews* 116 (4), 1803-1949
N,N-Diaryl Dihydrophenazines as Photoredox Catalysts for PET-RAFT and Sequential PET-RAFT/O-ATRP. *ACS macro letters* 7 (6), 662-666

Response: We have cited the two papers as Ref. 8 and Ref. 27 respectively in the revised manuscript (*Chem. Rev.* **2016**, *116*, 1803–1949; *ACS Macro Lett.* **2018**, *7*, 662–666). Thank you very much for the suggestion.

In conclusion, in my opinion, it is a strong study which deserves publication. The work is significant and new. I think it will have a significant impact in the polymer field.

Response: Thank you again for all the comments and suggestions.

Reviewer #2 (Remarks to the Author):

This manuscript reports the discovery of oxygen-doped antratherene as a new class of organic photoredox catalysts for metal-free ATRP. The reported catalyst features strong visible light absorption as well as good control over the whole polymerization process so that only low ppm levels of the catalyst was required to achieve the narrow molecular weight distributions. The most interesting part in this work is that the author rationalized the choice of oxygen doping in antratherene, leading to a superb catalyst in metal-free ATRP. The manuscript was well organized and presented, and I would like to recommend to publish this work after addressing several small points.

1. In table 1, DCM was used as a solvent in entry 3-14. In my knowledge, DCM is not a good solvent in ATRP since it could initiate the polymerization under ATRP conditions.

Response: Thank you very much for the comments and suggestions. The initiation by DCM should be negligible in the current system, as: 1) when the polymerization carried out in DCM in the absence of external initiators, almost no polymerization occurred (<5% conversion, please see Supplementary Table 4). Of note, this result could also suggest that the photocatalyst-initiation should be negligible; 2) In contrast to the quantitative (>98%) yield in the ATRA (atom-transfer radical addition) reaction of bromine initiator (diethyl 2-bromomalonate, DBM) with olefin using ODA catalyst (shown below), no ATRA product (no reaction) was detected by GC-MS analysis when dichloromethane was used instead of DBM. In fact, the reducing capability of the ODA photocatalysts (Figure 2, -1.6 to -1.8 V) is probably not enough to well reduce DCM (*J. Appl. Electrochem.* **1998**, *28*, 613–616).

2. In table 2, the columns of PC and initiator were shifted. For entry 15-17, CPADB was used as the initiator, so these reactions were not classified as ATRP but RAFT. It might be good to remove these examples.

Response: We have removed the results of RAFT polymerization (entries 15-17) from Table 2 in the revised manuscript. Thank you very much for the suggestion.

3. For almost all the examples, the obtained experimental Mn were about 50% more than theoretical values, which indicates a lower initiator efficiency. However, the dispersities are very low. The polymer sample should not be purified, and they should be directly analyzed by GPC to see the real Mn.

Response: To certain extent, the moderate initiator efficiency should be due to the initiators being used as received. Polymerization with freshly distilled initiators were also performed, which afforded much higher initiator efficiency (80-90%). For comparison, these results were placed in Supplementary Table 7 & 8 in the revised Supplementary Information. Thank you very much for this suggestion.

4. It would be better to cite a recent review of ATRP: *Chem. Soc. Rev.*, 2018, 47, 5457-5490.

Response: As suggested, we have cited this papers as Ref. 48 in the revised manuscript (*Chem. Soc. Rev.* 2018, 47, 5457-5490). Thank you again for all the comments and suggestion.

Reviewer #3 (Remarks to the Author):

Dear Authors

The studies developed in this work evaluates a catalyst design based on heteroatom-doping of polycyclic arenes as a new class of organic photoredox catalysts for O-ATRP.

I agree that this topic is of paramount importance, and believe that computationally speaking, the methodology employed in this work is sound and thus the results should be trustworthy. However minors revision is required before the paper can be considered for publication in Nature Communications. Below are the comments of the reviewer:

1. Define the acronyms the first time they appear in the text

Response: Thank you very much for the comments and suggestion. We have checked the manuscript carefully and defined the acronyms at the first time they appear in the text.

2. Review the English. I suggest making the text clearer.

Define all terms, abbreviations, acronyms mainly in the tables.

Response: Thank you very much for pointing out this. We have checked and revised the manuscript carefully, including several typos, confusing words, and the definition of some terms, abbreviations, acronyms in the main text and also in figures and tables. Thank you.

3. I suggest arranging the numerical order of the Figures in the manuscript. Figure 1a, Figure 1b, Figure 3c Example, figure 1c cannot appear before 1b. This is valid for every manuscript

Response: Thank you very much for pointing out this. We have adjusted the corresponding introduction part to fit the order of Figure 1a to 1d in the revised manuscript.

About the theoretical part:

4. Why did the authors use PBE0?

Response: PBE0 is a good functional for both geometrical optimization in the ground state for organic compounds and excitation energy (see: *J. Chem. Theory Comput.* **2016**, *12*, 459–465 and *J. Chem. Theory Comput.* **2018**, *14*, 3715–3727).

5. It was not indicated whether negative frequencies appeared. I suggest adding this information.

Response: As suggested, a sentence “No imaginary frequency was obtained at optimized geometries for all species.” was added in the computational method section in the revised Supplementary Information. Thank you very much for the suggestion.

6. To determine E₀ (PC+/3PC*) (Supl. Mat.) the authors used two reference values SHE (−4.48V) and to SCE (−0.244V) in acetonitrile. Indicate why of the values and why you use them.

Response: Cramer and Thuhlar’s paper (*J. Phys. Chem. B* **2006**, *111*, 408–422) supplies a reference value of −4.48V vs. SHE electrode for redox potential calculation in acetonitrile. To compare with the experimental results, the correction to SCE electrode of −0.244V was used.

7. Finally, I believe that TD-DFT calculations can bring more information to the paper. I suggest performing calculations at the TD-DFT level and correlating the results with the experimental data.

Response: We agree with the reviewer that TD-DFT can supply more insight to discover this reaction. Considering this communication is mainly focused on the discovery of the new and highly efficient catalysts, and the present calculations can support the experimental observation, we are also interested in the mechanism of this photocatalytic polymerization reaction. Thus, more TD-DFT calculations on the reactivity in the excited state and the whole reaction pathways are in plan. Thank you again for all the comments and suggestions.

REVIEWERS' COMMENTS

Reviewer #1 (Remarks to the Author):

I would like to thank the authors for answering my comments. I recommend publication.

Reviewer #2 (Remarks to the Author):

The author carefully addressed and answered all the points. I would recommend publishing this manuscript.

Reviewer #3 (Remarks to the Author):

Dear Author

Accepted the manuscript in its current form.

Point-by-point responses:

Reviewer #1 (Remarks to the Author):

I would like to thank the authors for answering my comments. I recommend publication.

Response: Thank you for your time and evaluation.

Reviewer #2 (Remarks to the Author):

The author carefully addressed and answered all the points. I would recommend publishing this manuscript.

Response: Thank you for your time and evaluation.

Reviewer #3 (Remarks to the Author):

Dear Author

Accepted the manuscript in its current form.

Response: Thank you for your time and evaluation.